# Rewiring of a KNOXI regulatory network mediated by UFO underlies the compound leaf development in *Medicago truncatula*

Zhichao Lu [1,3], Juanjuan Zhang[1,3], Hongfeng Wang[1,2,3], Ke Zhang[1], Zhiqun Gu[1], Yiteng Xu[1], Jing Zhang[1], Min Wang[1], Lu Han[1], Fengning Xiang[1] & Chuanen Zhou [1] ✉

*Class I KNOTTED-like homeobox* (*KNOXI*) genes are parts of the regulatory network that control the evolutionary diversification of leaf morphology. Their specific spatiotemporal expression patterns in developing leaves correlate with the degrees of leaf complexity between simple-leafed and compound-leafed species. However, *KNOXI* genes are not involved in compound leaf formation in several legume species. Here, we identify a pathway for dual repression of MtKNOXI function in *Medicago truncatula*. *PINNATE-LIKE PENTAFOLIATA1* (*PINNA1*) represses the expression of *MtKNOXI*, while PINNA1 interacts with MtKNOXI and sequesters it to the cytoplasm. Further investigations reveal that *UNUSUAL FLORAL ORGANS* (*MtUFO*) is the direct target of MtKNOXI, and mediates the transition from trifoliate to pinnate-like pentafoliate leaves. These data suggest a new layer of regulation for morphological diversity in compound-leafed species, in which the conserved regulators of floral development, *MtUFO*, and leaf development, *MtKNOXI*, are involved in variation of pinnate-like compound leaves in *M. truncatula*.

Angiosperm leaves exhibit great morphological diversity and are conventionally classified as simple or compound based on the number of blades on their petioles. A simple leaf consists of a single, undivided, continuous leaf blade, whereas a compound leaf consists of several discrete blade units termed leaflets. Leaf development follows a common developmental program, which can be flexibly adjusted species-specifically in a spatiotemporal manner[1]. Morphogenesis and differentiation are two key stages that determine the final leaf form in leaf development. Compared to simple leaf, compound leaf formation requires a prolonged morphogenetic phase that allows for the initiation of leaflets and specification of the midrib, petiole, and rachis[2]. This extended morphogenetic phase is enabled by the sufficient maintenance and modulation of the transient morphogenetic activity at specific leaf margin meristematic regions termed marginal blastozones[2]. Thus, maintaining a transient morphogenetic window

during leaf morphogenesis is crucial for the formation of compound leaves.

The *KNOTTED-like homeobox* (*KNOX*) and *BEL1-like homeobox* (*BLH*) genes constitute the plant-specific three-amino-acid loop extension (TALE) superfamily, which plays a crucial role in plant development[3]. The KNOX and BELL proteins form heterodimers that are implicated in nuclear localization and regulate DNA-binding affinity[4–7]. *KNOX* genes can be further categorized into three subclasses, *Class I KNOX*, *Class II KNOX*, and *Class M KNOX*[8]. *Class I KNOX* (*KNOXI*) transcription factors are essential for maintaining the shoot apical meristem (SAM) activity and regulating leaf complexity in angiosperms[9–12]. In most compound-leafed species, including tomato (*Solanum lycopersicum*) and hairy bittercress (*Cardamine hirsuta*), KNOXI proteins are crucial for maintaining transient morphogenetic activity during leaf development, as their reactivation in the leaf

[1]The Key Laboratory of Plant Development and Environmental Adaptation Biology, Ministry of Education, School of Life Sciences, Shandong University, Qingdao 266237, China. [2]Shandong Peanut Research Institute, Qingdao 266199, China. [3]These authors contributed equally: Zhichao Lu, Juanjuan Zhang, Hongfeng Wang. ✉e-mail: czhou@sdu.edu.cn

primordium induces leaflet initiation[13,14]. Furthermore, *KNOXI* has been thought to promote morphogenesis in most compound-leafed species, and its overexpression dramatically increases the number of leaflets. In tomato, the BLH protein BIPINNATE (BIP) interacts with the KNOXI protein to form a heterodimer that is re-located to the nucleus, resulting in determinate growth for leaf morphogenesis[15]. The loss-of-function of BIP leads to increased primary and secondary leaflet production and higher expression of *KNOXI*. In legume plants belonging to the inverted repeat-lacking clade (IRLC), such as pea and *Medicago truncatula*, KNOXI seems to be not involved in compound leaf development. The expression of *KNOXI* is not reactivated in developing leaf primordia, as distinguished from most compound-leafed species[16,17]. The role of KNOXI is replaced by the transcription factor *LEAFY* (*LFY*) orthologues, *UNIFOLIATA* (*UNI*) in pea and *SINGLE LEAFLET1* (*SGL1*) in *M. truncatula*, which are essential for regulating the transient morphogenetic window of leaf growth[18–20]. The loss-of-function *sgl1* mutant fails to generate trifoliate adult leaves but produces a single blade. However, the constitutive expression of *SGL1* did not increase leaf complexity, which is distinguished from the overexpression of *STM/BP-like KNOXI* expression in *M. truncatula*, indicating that *SGL1* and *MtKNOXI* act in parallel pathways to regulate different targets[17].

*Medicago truncatula* is a model legume with leaves having a typical trifoliate pattern. Recent findings have revealed the regulatory mechanisms of compound leaf patterning in *M. truncatula*, for instance, the BLH protein PINNATE-LIKE PENTAFOLIATA1 (PINNA1) represses the transcription of *SGL1* in a spatiotemporal manner to define the trifoliolate form in wild-type plants[21]. However, the leaf pattern in *SGL1*-overexpressing plants is similar to that in wild-type plants[17], implying that the increased activity of *SGL1* is not responsible for the generation of ectopic leaflets in the *pinna1* mutant. Here, we characterized a novel regulatory pathway modulated by *PINNA1* for the formation of additional leaflets in pinnate-like compound leaves in *M. truncatula*. We found that mutations in *PINNA1* led to the formation of ectopic leaflets in pinnately compound leaves, and the defects in compound leaf patterning in *pinna1* could be partially rescued by the simultaneous disruption of the *STM/BP-like* and *KNAT2/6-like MtKNOXI* genes. Genetic and biochemical analysis revealed that PINNA1 forms a heteromeric complex with MtKNOXI and sequesters it to the cytoplasm, preventing the direct activation of the target gene, *UNUSUAL FLORAL ORGANS* (*UFO*) ortholog *MtUFO*, whose ectopic expression leads to a pinnate-like form with five leaflets. In this work, we proposed a model to understand how trifoliate leaves transform into pinnate-like pentafoliate leaves and shed light on the regulatory mechanisms of compound leaf formation in *M. truncatula*.

## Results

### Comprehensive characterization of the roles of *Class I MtKNOX* in compound leaf development

Phylogenetic analysis revealed that the *STM*-like genes (*MtKNOX1* and *MtKNOX6*), a *BP*-like gene (*MtKNOX2*), and a *KNAT2/6*-like gene (*MtKNOX7*) are grouped into *Class I KNOX* genes (*MtKNOXI*) in *M. truncatula* (Fig. 1a). Additionally, we found that *MtKNOX8* and its homologs in legume species formed a distinct clade which is not grouped with other *MtKNOXI* genes (Fig. 1a and Supplementary Fig. 1a). Multiple sequence alignment analysis showed that the N-terminus region of the MtKNOX8 protein had a shorter amino acid sequence compared to other KNOXI proteins (Supplementary Fig. 2). This result implies that the function of MtKNOX8 is different from other members of the MtKNOXI proteins. In compound-leafed species, such as *S. lycopersicum* and *C. hirsuta*, the ectopic expression of *KNOXI* genes results in a dramatic increase in leaf complexity[3,13,22,23]. To comprehensively assess if *MtKNOX1/2/6/7* genes have conserved roles, they were introduced into wild-type

plants under the control of the *CaMV35S* promoter. The resulting transgenic plants displayed a similar phenotype that the reiteration of higher-order leaflets was produced along petiolules (Supplementary Fig. 1b–j), but those plants had a seedling-lethal phenotype. The transgenic plants with relatively low expression of *MtKNOXI* could survive, and showed a weak phenotype with two ectopic leaflets formed in a pinnately compound leaf form (Fig. 1b–f and Supplementary Fig. 1g–j). We also assessed the function of *MtKNOX8*; however, the *35S:MtKNOX8* transgenic plants showed no obvious changes in leaf morphology (Fig. 1g and Supplementary Fig. 1k), suggesting distinct roles of MtKNOX8 and MtKNOX1,2,6,7 proteins. These observations indicate that ectopic *MtKNOXI* activity is sufficient for increasing leaf complexity.

To better understand the functions of *MtKNOXI* genes, their loss-of-function mutants were investigated. A previous study showed that the simultaneous disruption of *STM/BP*-like *MtKNOXI* genes did not lead to obvious defects in leaf morphology[17]. To determine whether the *KNAT2/6*-like gene is involved in compound leaf development, two mutant alleles of *MtKNOX7* were obtained by the reverse genetic screening of the *Tnt1*-tagged mutant population (Fig. 1h). A reverse transcription-PCR showed no detectable *MtKNOX7* transcripts in mutants (Fig. 1i), and the *mtknox7* mutant showed no obvious defects in leaf morphology (Fig. 1j–l). To further assess functional redundancy between *STM/BP*-like and *KNAT2/6*-like *MtKNOXI* genes, a quadruple mutant was generated. Similar to the *mtknox1/2/6* triple mutant (Fig. 1m), knockout of *MtKNOX1/2/6/7* in *M. truncatula* did not affect leaf development (Fig. 1n), indicating that *MtKNOXI* genes are not involved in compound leaf patterning.

### MtKNOX1/2/6/7 and PINNA1 interact physically

Ectopic expression of *MtKNOXI* led to enhanced morphogenetic activity in developing compound leaves, implying that the function of *MtKNOXI* was repressed during leaf primordia development in *M. truncatula*. To identify a possible repressor, we used the full-length *MtKNOX7* cDNA as a bait to identify potential interacting partners by a yeast two-hybrid screening. We found that MtKNOX7 interacted strongly with a BLH protein that was reported previously, PINNA1[21]. Further analysis of the yeast two-hybrid X-a-Gal filter assay showed that MtKNOX1/2/6/7 were also able to interact with PINNA1 (Supplementary Fig. 3). To verify these interactions in vivo, a bimolecular fluorescence complementation (BiFC) experiment and a coimmunoprecipitation (co-IP) assay were carried out in *Nicotiana benthamiana*. For the BiFC assay, the MtKNOX1/2/6/7 proteins were fused with the N-terminus of YFP (YN), and PINNA1 was fused with the C-terminus of YFP (YC). A co-expression of PINNA1-YC and each MtKNOX1/2/6/7-YN resulted in significant YFP signals in the cytoplasmic space of leaf epidermal cells, suggesting their strong extranuclear interaction (Fig. 2a and Supplementary Fig. 4). Furthermore, these YFP signals were found to co-localize with the puncta of the autophagosome marker mCherry-ATG8e and the autophagy receptor NBR1-mCherry[24,25] (Supplementary Fig. 4a, b). This observation indicates that the complexes were localized within autophagosomes or autophagic bodies and could potentially undergo degradation through the autophagy process. Additionally, the co-IP assays showed that PINNA1-7Myc could be coimmunoprecipitated with MtKNOX1/2/6/7-GFP (Fig. 2b), suggesting that PINNA1 can heterodimerize with MtKNOX1/2/6/7. Previous studies indicate that BLH proteins heterodimerize with KNOX proteins through their KNOX MEINOX domain[26,27], which is composed of two subdomains, MEINOX1 (also referred to as KNOX1) and MEINOX2 (also referred to as KNOX2), essential for BLH-KNOX dimerization. To identify which MEINOX domain interacted with PINNA1, a yeast two-hybrid assay was performed. The data showed that MEINOX1 was sufficient for the interaction between MtKNOX7 and PINNA1, and MEINOX2 was sufficient for

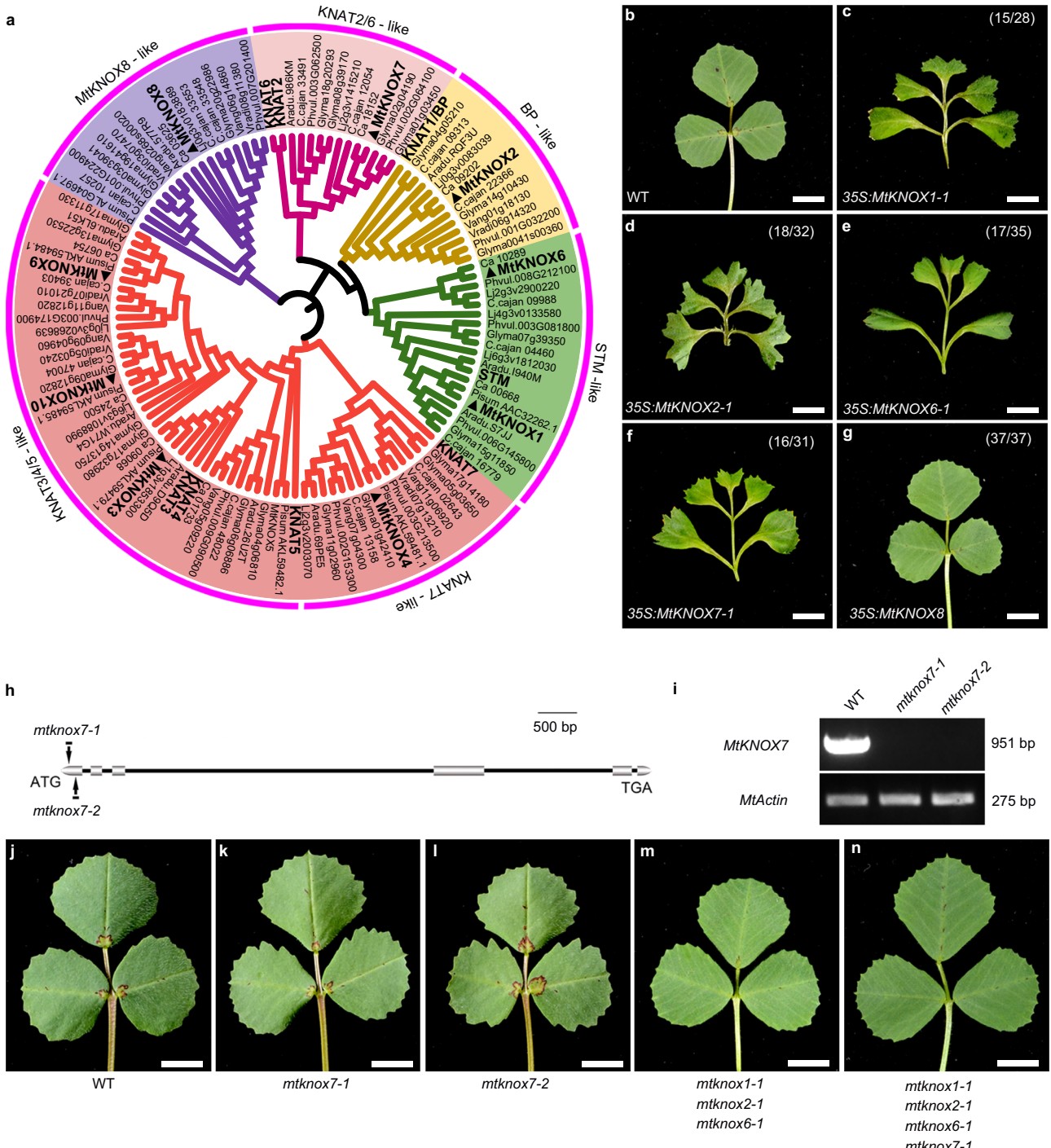

**Fig. 1 | Comprehensive characterization of the roles of *MtKNOXI* in compound leaf development. a** Phylogenetic tree of KNOX proteins from several angiosperms (*A. thaliana, M. truncatula, C. cajan, G.max, L. japonicas, C. arietinum, P. vulgaris, A. duranensis, V. angularis,* and *V. radiata,* and *P. sativum*). **b** Leaf morphology of wild-type (WT) plants. **c**–**f** Representative leaves derived from transgenic plants over-expressing *35S:MtKNOXI* (**c**), *35S:MtKNOX2* (**d**), *35S:MtKNOX6* (**e**), and *35S:MtKNOX7* (**f**) with a weak phenotype, respectively. **g** Mature leaf in transgenic plants over-expressing *35S:MtKNOX8*. (*w/o*) in (**c**–**g**) indicates that *w* in *o* total transgenic plants shows the displayed features. **h** Gene model of *MtKNOX7* and *Tnt1* insertion positions in *mtknox7* alleles. Boxes represent exons and lines represent introns. **i** Reverse transcription-PCR (RT-PCR) analysis of *MtKNOX7* transcripts in WT and *mtknox7* mutants. *MtActin* was used as the loading control. Similar results were obtained from three independent experiments. **j**–**n** Representative leaves of WT (**j**), *mtknox7-1* (**k**), *mtknox7-2* (**l**), *mtknox1-1 mtknox2-1 mtknox6-1* triple mutants (**m**), and *mtknox1-1 mtknox2-1 mtknox6-1 mtknox7-1* quadruple mutants (**n**). Scale bars, 5 mm in (**b**–**g**) and (**j**–**n**). Source data are provided as a Source Data file.

the interaction between MtKNOX1/2/6 and PINNA1 (Supplementary Fig. 5). Such different interaction modes between MtKNOX1/2/6-PINNA1 and MtKNOX7-PINNA1 likely suggested a functional difference between the *STM/BP*-like and *KNAT2/6*-like *MtKNOXI* genes, as the domains of a protein regulate its functions and interplays[28].

## MtKNOXI activities partially contribute to defects in compound leaf patterning in *pinna1* mutants

To characterize the relationship between MtKNOXI and PINNA1, we isolated three new *pinna1* knockout mutant alleles by reverse genetic screening (Fig. 2c and Supplementary Fig. 6a). Approximately 89% of

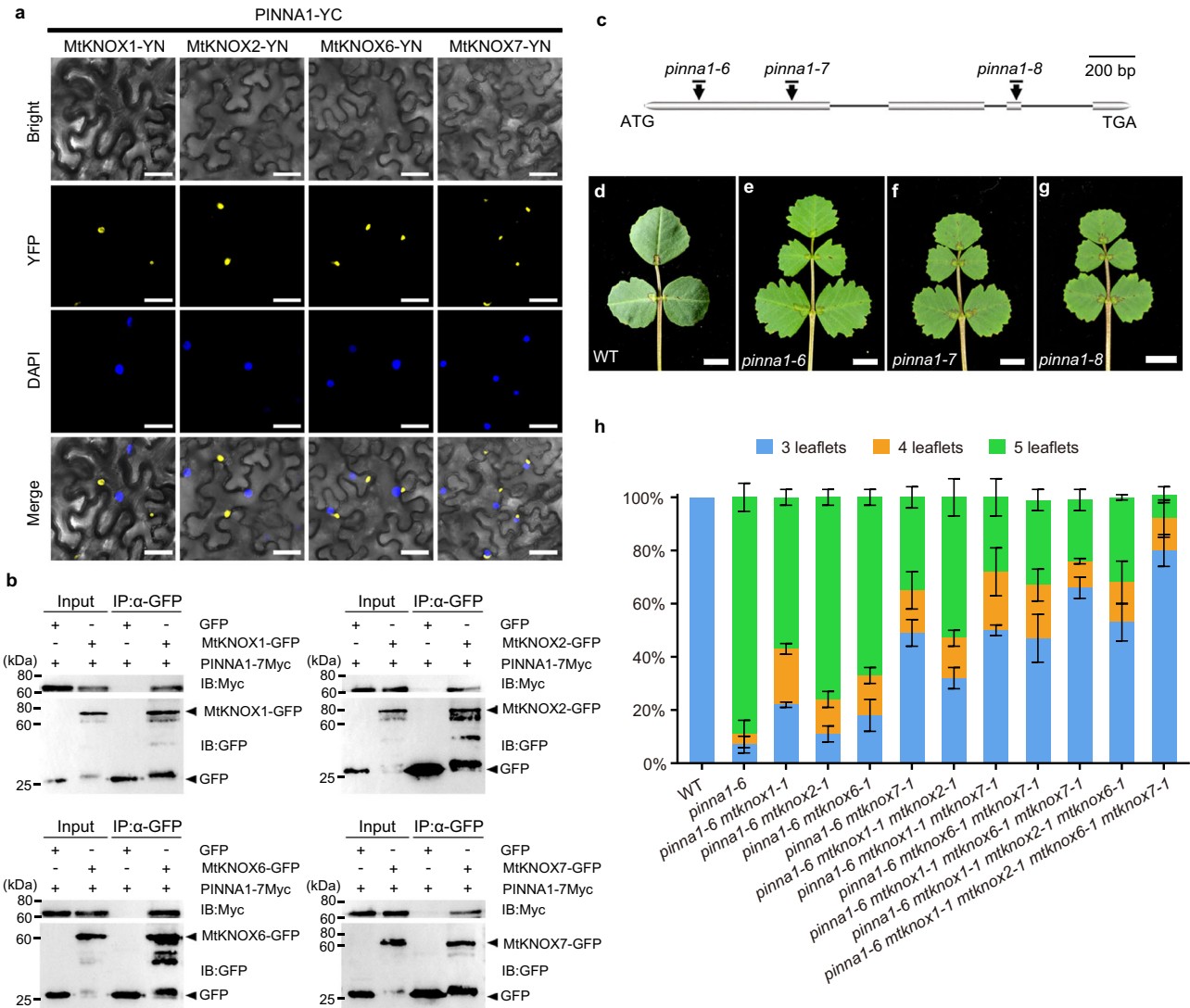

**Fig. 2 | Physical and genetic interaction of MtKNOXI and PINNA1. a** BiFC assay was performed in tobacco epidermal cells, and it was observed that PINNA1 interacts with MtKNOX1, MtKNOX2, MtKNOX6, and MtKNOX7, as indicated by the presence of a yellow fluorescent signal. DAPI (4',6-diamidino-2-phenylindole) was used to stain the nuclei. Similar results were obtained from three independent experiments. **b** Coimmunoprecipitation assay confirmed that PINNA1 interacts with MtKNOX1/2/6/7 in tobacco epidermal cells. Total proteins were immunoprecipitated using GFP-Trap beads, and the coimmunoprecipitated proteins were detected by the anti-Myc antibody. Three biological repeats were performed for each interaction. **c** Schematic diagram of *PINNA1* gene structure and *Tnt1* insertions in *pinna1* alleles. Boxes represent exons and lines represent introns. **d** Representative leaves of WT. **e–g** Representative leaves of three *pinna1* mutants, *pinna1-6* (**e**), *pinna1-7* (**f**), and *pinna1-8* (**g**). **h** Proportion of leaves with different leaflet numbers in WT, *pinna1-6* single mutant, and *pinna1-6 mtknoxi* multiple mutants. Fifty-day-old plants were used for counting the leaflet numbers of adult leaves at least three biologically independent replicates. Data represent means ± SD (*n* = 3 biological replicates). Scale bars, 20 µm in (**a**) and 5 mm in (**d–g**). Source data are provided as a Source Data file.

adult leaves in *pinna1* produced five leaflets in a pinnately compound leaf form (Fig. 2d–g and Supplementary Fig. 6b–d). A scanning electron microscopy (SEM) analysis showed that a pair of incipient ectopic leaflets of *pinna1-6* emerged as two bulging cell groups from the proximal zones of the terminal leaflet primordium (Supplementary Fig. 6e–h). It is noteworthy that the compound leaf pattern in *pinna1* was similar to that of *35S:MtKNOX1/2/6/7* plants with a weak phenotype (Fig. 1c–f). These observations implied that the MtKNOX1/2/6/7 activity may contribute to the formation of ectopic leaflets in *pinna1* mutants. To validate this hypothesis, we generated double, triple, quadruple, and quintuple mutants among the *mtknoxi* and *pinna1* mutants. Among different double mutant combinations, approximately 49% of leaves in *pinna1-6 mtknox7-1* were trifoliate, but only 10.9 to 21.8% of leaves were rescued in the other three double mutant combinations (Fig. 2h). Moreover, knockout of all *MtKNOX1, 2, 6* in *pinna1-6* could rescue 53.2% of leaves (Fig. 2h), which was similar to that in *pinna1-6*

*mtknox7-1*, indicating a comparable contribution of *MtKNOX1, 2, 6* and *MtKNOX7* to the defects in *pinna1-6*. Furthermore, knockout of all *MtKNOXI* genes in *pinna1-6* rescued 79.8% of leaves of the mutant (Fig. 2h), indicating the redundant roles of *STM/BP*-like and *KNAT2/6*-like *MtKNOXI* in the regulation of compound leaf patterning in *pinna1-6* mutants. Taken together, this genetic evidence suggests that the leaf phenotype seen in *pinna1* is caused partially by the activities of *MtKNOXI*.

**PINNA1 mainly represses the activities of MtKNOXI by regulating its subcellular localization**
To our knowledge, BLH-KNOX heterodimerization is functional to the translocation of the BLH-KNOX complex from the cytoplasm to the nucleus[4–7]. However, all PINNA1-MtKNOXI complexes were not trafficked into the nucleus in this study (Fig. 2a), indicating the existence of a distinct BLH-KNOX regulatory circuitry in *M. truncatula*.

To investigate the underlying possible reasons for this, the subcellular localizations of these proteins were first determined. In plants, the relative accessibility of nuclear export signals (NES) and/or nuclear localization signals (NLS) generally determines the nucleo-cytoplasmic partitioning of proteins[29]. We found that PINNA1 and all MtKNOX1/2/6/7 proteins have conserved amino acids that match the NES amino acid sequence (Supplementary Fig. 7), and speculated that they are not localized in the nucleus. In fact, both PINNA1-GFP and PINNA1-mCherry fusions were excluded from the nucleus in *M. truncatula* protoplasts (Fig. 3a, b and Supplementary Fig. 8a). This finding is distinctly different from a previous report that PINNA1 was localized to the nucleus in tobacco leaf epidermal cell[21]. We also transiently expressed PINNA1-GFP or GFP-PINNA1 in *M. truncatula* epidermal cells, and both proteins were found localized to the cytoplasmic space (Supplementary Fig. 8b). To further confirm these results, we performed a nuclear-cytoplasmic fractionation assay to test the localization of GFP- or Myc-tagged PINNA1 proteins by immunoblotting the nuclear and cytoplasmic fractions of transfected *M. truncatula* protoplasts. The immunoblot analysis showed that both GFP- and Myc-tagged PINNA1 proteins were mostly present in the cytoplasmic fraction (Supplementary Fig. 8c). Therefore, it is unlikely that PINNA1 functions as a transcriptional factor to regulate the downstream genes. Next, we determined the subcellular localization of MtKNOX1/2/6/7-GFP. We found that less than 10% of GFP signals of each MtKNOX1/2/6-GFP were nucleo-cytoplasmic; however, MtKNOX7-GFP showed 100% nucleo-cytoplasmic distribution (Fig. 3a, b and Supplementary Fig. 8d), indicating that different regulation mechanisms are mediated by MtKNOX7.

To further understand the biological effects of PINNA1-MtKNOXI heterodimerization, we transiently co-expressed each MtKNOX1/2/6/7-GFP with PINNA1-mCherry in protoplasts. The data showed that all GFP/mCherry signals were excluded from the nucleus (Fig. 3a, c). As the control, 35S-mCherry did not affect the subcellular localization of each MtKNOX1/2/6/7-GFP protein (Fig. 3a, c). These results indicated that PINNA1 prevents the nuclear localization of MtKNOX1/2/6/7 from promoting leaflet development. To further confirm this, a dual-luciferase reporter (DLR) system assay was performed by transiently expressing the fusion protein GAL4BD-MtKNOX1/2/6/7 with *6XUA-Spro:LUC* reporter in protoplasts. MtKNOX1/2/6 displayed relatively low transactivation activity, except MtKNOX7 (Fig. 3d and Supplementary Fig. 8e), consistent with their different subcellular localization patterns. Subsequently, each of the GAL4BD-MtKNOXI members and *35S:PINNA1* were co-expressed in protoplasts along with the *6XUA-Spro:LUC* reporter, and it was found that the activity of MtKNOX7 was repressed by PINNA1 (Fig. 3d and Supplementary Fig. 8e). The activities of MtKNOX1/2/6 were not altered significantly by PINNA1, probably because of their mostly cytoplasmic localization. Previous reports showed that the KNOXI expression is affected in the mutants of the *BLH* gene, which is involved in the generation of leaf-form variations[15,30]. We also found that the transcriptional levels of *MtKNOXI* were slightly increased in the *pinna1* mutant (Supplementary Fig. 9). To further investigate the relationship between *MtKNOX7* and *PINNA1*, the expression patterns of *MtKNOX7* and *PINNA1* were examined by RNA in situ hybridization analysis. The results showed that *MtKNOX7* transcripts were only present in shoot apical meristem (SAM), but not in the incipient leaf primordia (S0) and developing leaf primordia in the wild type (Fig. 3e). However, the spatial localizations of *PINNA1* were detected in SAM, S0, S1, and developing leaf primordia in the wild type (Fig. 3f). Thus, *MtKNOX7* and *PINNA1* have the overlapped expression domain in the SAM and the boundary between SAM and the emerging leaf primordia (Fig. 3g), suggesting that there is a potential for the interaction between them. Additionally, the ectopic expression signal of *MtKNOX7* was detectable in the S0, S1, and S2 primordia in the *pinna1-6* mutants (Fig. 3h). These data suggest that the leaf phenotype seen in *pinna1* is probably caused, at least partially,

by the ectopic expression of *MtKNOX7*. Taken together, PINNA1 has diverse inhibitory effects on MtKNOXI (Fig. 3i), which are mainly mediated by the regulation of nucleo-cytoplasmic partitioning and repression of the expression domain of MtKNOX7.

## Pentafoliate leaves in *pinna1* result from the increased activity of *MtUFO*

In *M. truncatula*, all adult leaves in *sgl1* mutant were simple[18], indicating that *SGL1* is necessary for regulating indeterminacy during leaf development. The role of *PINNA1* in producing lateral leaflets was further examined by generating double mutants with the *sgl1* mutant. Most leaves in the *sgl1 pinna1* double mutant were simple (Supplementary Fig. 10a–f), suggesting that *sgl1* is genetically epistatic to *pinna1* in leaf patterning. In our previous report, however, leaf complexity was not changed in *SGL1*-overexpressing plants (Supplementary Fig. 10g, h)[17], indicating that the ectopic *SGL1* activity is not sufficient for increasing leaf complexity. Therefore, the generation of the two extra leaflets in *pinna1* was not due to the ectopic expression of *SGL1*. To determine the potential regulator of the defects in *pinna1*, we identified the genes involved in the regulatory network of *LFY/SGL1*. It has been shown that LFY functions with its coregulatory gene *UNUSUAL FLORAL ORGANS* (*UFO*) in flower development[31]. In addition, the ectopic expression of *UFO* in *Arabidopsis* and *C. hirsuta* led to deeply serrated leaf margins and increased leaf complexity, respectively[31,32]. These observations indicate that *UFO* promotes indeterminacy in leaf development, implying that UFO orthologs may play an important role in compound leaf patterning in *M. truncatula*. To test this hypothesis, we overexpressed *MtUFO* under the *CaMV35S* promoter. The transgenic plants exhibited five leaflets in a pinnate-like form, similar to those in *pinna1* (Fig. 4a–c) and *35S:MtKNOXI* plants with weak phenotype (Fig. 1c–f). This result indicated that the ectopic *MtUFO* activity promotes morphogenetic activity during leaf morphogenesis in *M. truncatula*. To investigate the role of *MtUFO* in the development of pentafoliate leaves in *pinna1*, we compared the expression levels of *MtUFO* in WT and *pinna1*. Our qRT-PCR analysis revealed a significant upregulation of *MtUFO* expression in *pinna1* (Fig. 4d). To further explore the function of *MtUFO*, a reverse genetic screening of the *Tnt1*-tagged mutant population was performed (Supplementary Fig. 11a–c). In total, four loss-of-function mutants of *MtUFO* were isolated, all of which produced normal trifoliate leaves (Fig. 4e). Subsequently, *mtufo-1* was crossed with *pinna1-6* to generate the double mutant, and the leaf defects in *pinna1-6* were fully rescued in *pinna1-6 mtufo-1* (Fig. 4f). Furthermore, RNA in situ hybridization experiments showed that the expression domain of *MtUFO* was expanded in the leaf primordium (S1) of the *pinna1-6* mutant, in comparison to the WT (Supplementary Fig. 11d, e). These observations indicate that the increased activity of *MtUFO* is responsible for the formation of extra leaflets in *pinna1*.

## *MtUFO* functions downstream of the PINNA1-MtKNOX7 complex

To assess the effects of PINNA1 or MtKNOXI on the expression of *MtUFO*, a DLR assay was performed using the GFP-tagged PINNA1 or MtKNOXI effectors. PINNA1 and MtKNOX1/2/6 had no influence on the expression of LUC driven by a 2.9-kb promoter of *MtUFO*; however, it was effectively activated by MtKNOX7 semi-in vivo (Supplementary Fig. 12). In addition, the *MtUFO* promoter-driven LUC expression was significantly activated by GR-tagged MtKNOX7 upon treatment with dexamethasone (DEX) (Fig. 5a, b). The KNOXI protein is reported to bind to DNA sequences containing tandem TGAC core motifs[33–37]. Electrophoretic mobility shift assay (EMSA) revealed that the *MtUFO* promoter contains the cis-element TGACTTGAC, which acts as a binding site for MtKNOX7, at 1974 bp upstream of the translation initiation codon (Fig. 5c). The binding of MtKNOX7 to the TGACTTGAC

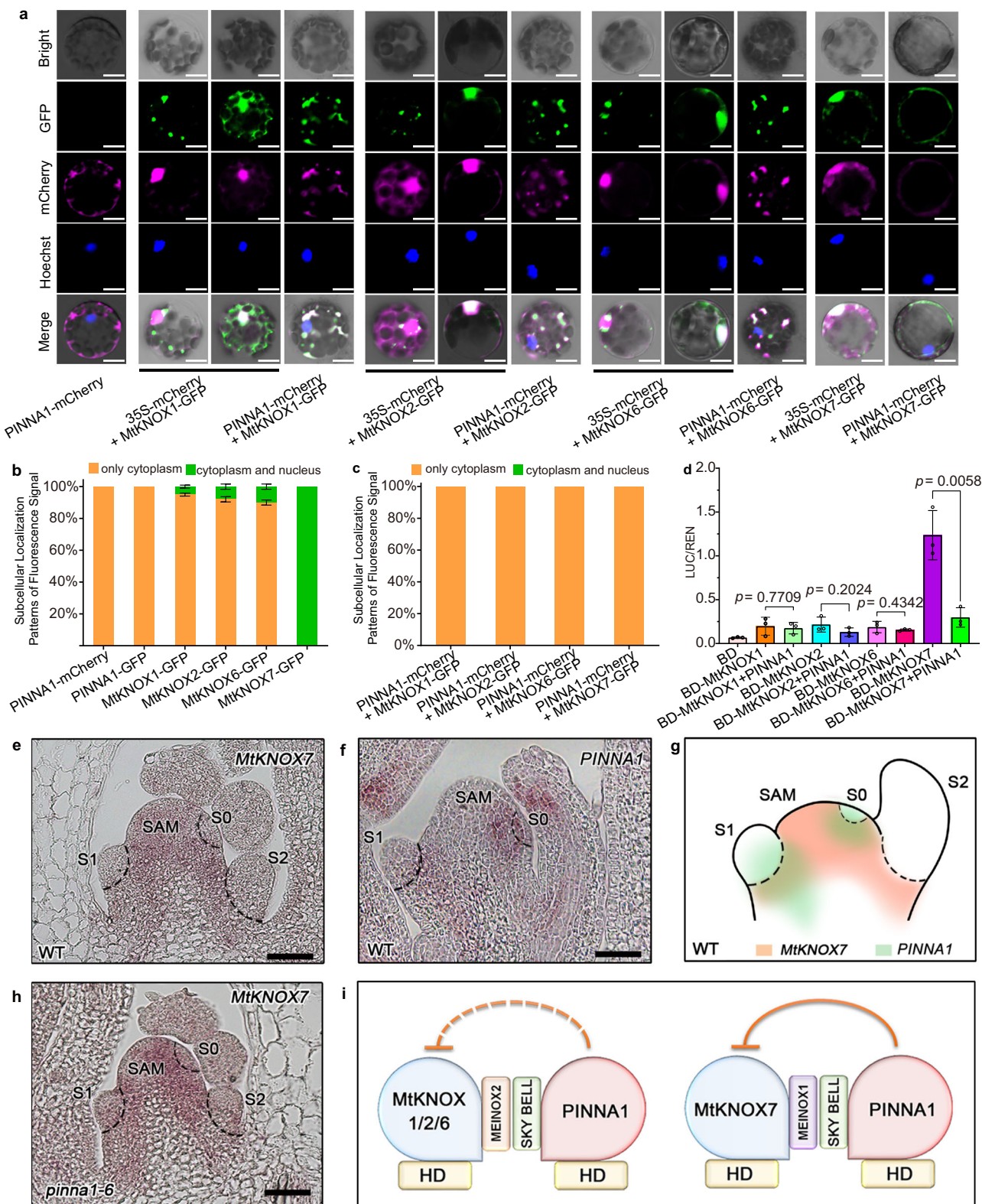

cis-element was also confirmed by chromatin immunoprecipitation (ChIP) assay. A ChIP-qPCR analysis showed that the *MtUFO* promoter region containing the TGACTTGAC cis-element was significantly enriched in the precipitated DNA from *35S:MtKNOX7-GFP* plants. (Fig. 5d). Moreover, to test the effect of the PINNA1-MtKNOX7 complex on the activities of the *MtUFO* promoter (*proMtUFO*), we performed a DLR assay. The activity of *proMtUFO* co-expressed with PINNA1 and MtKNOX7 was effectively repressed compared to when it was activated by MtKNOX7 alone, indicating that PINNA1 represses the expression of *MtUFO* by interacting with MtKNOX7 (Fig. 5e). To further confirm this, the expression levels of *MtUFO* were measured and found significantly upregulated in *35S:MtKNOX7* plants, compared to wild-type plants (Fig. 5f). In addition, the ectopic leaflets in *35S:MtKNOX7* plants were completely recovered when *MtUFO* was knocked out (Fig. 5g, h). These data suggested that *MtUFO* functions downstream of the PINNA1-MtKNOX7 complex for promoting indeterminacy in leaf development.

**Fig. 3 | PINNA1 regulates the subcellular localization of MtKNOXI and represses their activities. a** Co-localization of PINNA1-mCherry and each MtKNOX1/2/6/7-GFP in the *Medicago* protoplasts. Nuclei were stained by Hoechst (Hoechst 33342). **b** Quantification of subcellular localization patterns of PINNA1-mCherry, PINNA1-GFP, and each MtKNOX1/2/6/7-GFP in protoplasts respectively. **c** Quantification of subcellular localization patterns of GFP/mCherry fluorescence signal when expressed PINNA1-mCherry together with each MtKNOX1/2/6/7-GFP in protoplasts. Ratios of subcellular localization patterns were calculated from 50 protoplasts with at least three biologically independent replicates in (**b**) and (**c**). Data represent means ± SD (*n* = 3 biological replicates). **d** Transcriptional activation assay in *Arabidopsis* protoplast indicated that PINNA1 represses the activity of MtKNOX7. Data represent means ± SD (*n* = 3 biological replicates) and *P* values were calculated by

unpaired two-tailed *t*-test. **e** RNA in situ hybridization of *MtKNOX7* in the leaf primordia of WT. **f** RNA in situ hybridization of *PINNA1* in the leaf primordia of WT. **g** Schematic representations of expression patterns of *MtKNOX7* and *PINNA1* in the leaf primordia of WT. **h** RNA in situ hybridization of *MtKNOX7* in the leaf primordia of *pinna1-6*. **i** Schematic representations of the inhibitory effect of PINNA1 on the functions of MtKNOX1/2/6/7. PINNA1 has stronger inhibitory effects on MtKNOX7 than that on MtKNOX1/2/6. The orange full curve represents the stronger inhibitory effects and the orange dotted curve represents the weaker ones. The subcellular localization experiments in (**a**) and the RNA in situ hybridization in (**e**, **f**, **h**) were independently repeated three times with similar results. Scale bars, 5 μm in (**a**), and 50 μm in (**e**, **f**, **h**). Source data are provided as a Source Data file.

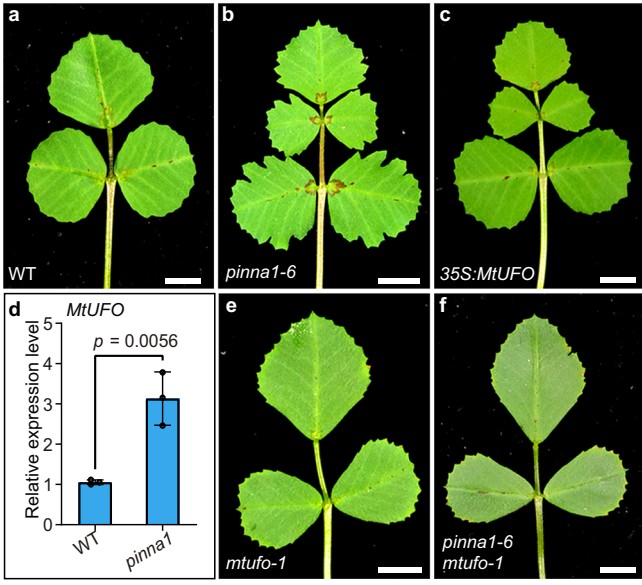

**Fig. 4 | Pentafoliate leaves in *pinna1* result from the increased activity of *MtUFO*. a–c** Representative leaves of WT (**a**), *pinna1-6* (**b**), and *3SS: MtUFO* (**c**). **d** The expression levels of *MtUFO* in the shoot apices of WT and *pinna1* were determined by qRT-PCR. *MtUBIQUITIN* was used as the internal control. Data represent mean ± SD (*n* = 3 biological replicates) and *P* values were calculated by unpaired two-tailed *t*-test. **e, f** Representative leaves of *mtufo-1* (**e**) and *pinna1-6 mtufo-1* double mutants (**f**). Scale bars, 5 mm in (**a–c**, **e**, **f**). Source data are provided as a Source Data file.

## Discussion

Several studies have shown that interactions between TALE (three-amino acid loop extension) homeodomain proteins play important roles during plant and animal development[26,38]. In animals, the interactions between the TALE proteins MEIS and PBC mask the NES in the PBC protein from the nuclear export receptor CRM1/exportin-1, thereby facilitating the nuclear localization of the MEIS-PBC heterodimers[39]. In plant development, the combinatorial interactions among TALE proteins make up functional BLH-KNOX heterodimers[40]. Selective BLH-KNOX interactions guide the correct subcellular localization of BLH-KNOX complexes and mediate DNA-binding activity, which plays distinct roles in plant growth and development. Here, our data suggest conserved and diverged molecular mechanisms by which the functional BLH-KNOX complexes control compound leaf development in *M. truncatula*. Our data showed that the BLH protein PINNA1 heterodimerizes with STM/BP-like and KNAT2/6-like MtKNOX in the cytoplasmic space, in contrast to the previously reported nuclear localization of the BLH-KNOX complexes[4–7]. In particular, the cytoplasm-localized PINNA1-MtKNOX7 heterodimers prevent MtKNOX7 protein trafficking to the nucleus, which inhibits transcriptional regulation of the target genes controlling compound leaf development (Fig. 6).

Nucleo-cytoplasmic transport is an important means of regulating the subcellular localization of various TALE proteins, which is important for their proper functioning. In TALE transcription factors, NES and NLS are generally short peptides recognized by a variety of nuclear importin and exportin proteins required for nucleus-cytoplasm trafficking[41]. In plants, the nuclear import mechanism of the BLH-KNOX complexes is similar to that of the PBC-MEIS complex in animals. The heterodimerization of BLH-KNOX likely masks the NES to preclude the recognition of CRM1, thus causing BLH-KNOX heterodimers to accumulate in the nucleus[6]. Therefore, the nuclear import of BLH-KNOX complexes is correlated to the effect of masking the NES by heterodimerization and balance between the activities of the NES and NLS. In this study, we used NLStradamus software (www.moseslab.csb.utoronto.ca/NLStradamus/) to predict the NLS of PINNA1 and MtKNOXI. All MtKNOXI proteins had a predictable NLS (Supplementary Fig. 13), but PINNA1 did not, suggesting that the cytoplasmic localization of PINNA1 was probably due to the lack of efficient NLS. Moreover, all PINNA1-MtKNOXI complexes were localized in the extranuclear space, probably because the heterodimerization of PINNA1-MtKNOXI did not mask the NES that was recognized by CRM1 or because the heterodimers lacked an efficient NLS. Therefore, treatment with leptomycin B (LMB) specifically inhibits the CRM1 activity[42] or the fusion of an efficient NLS to PINNA1, thus providing an insight into the mechanism of the nuclear export of PINNA1-MtKNOXI complexes in the future.

The subcellular localization diversity existing among KNOXI orthologs in *M. truncatula* has also been observed in other species. In rice, three KNOXI (*Oskn1/OSH1*, *Oskn2/OSH71*, and *Oskn3/OSH15*) proteins showed different nuclear and cytoplasmic localization patterns. Different from the only nucleo-cytoplasmic localization of Oskn2 or Oskn3, the localization of Oskn1 varied between different cells, showing either cytoplasmic or nucleo-cytoplasmic distribution[43]. The subcellular localization of KNOXI was influenced by plant hormones. For example, the application of gibberellic acid (GA3) and 1-naphthylacetic acid (1-NAA) completely changed the localization of all Oskn1-3 proteins from the nucleus to the cytoplasm, whereas the application of naphthylphthalamic acid (NPA) led to their nuclear concentration[43]. Moreover, the subcellular localization of KNOXI might be influenced by other proteins, such as the members of the multiple C2 domain and transmembrane region protein (MCTP) family and a microtubule-associated protein. The MCTP protein FTIP3/4 prevents intracellular trafficking of STM from endosomes to the plasma membrane and facilitates STM recycling to the nucleus but does not affect KNAT2 in *Arabidopsis*[44], and OsFTIP7 also mediates the nucleo-cytoplasmic translocation of OSH1 in rice[45]. MPB2C, a microtubule-associated protein, can alter the subcellular distribution of KNOXI protein KN1[46]. Therefore, different subcellular localizations between MtKNOX1/2/6 and MtKNOX7 in single cells could be due to multiple regulatory mechanisms.

MtKNOX8 and its homologs in legume species formed a distinct clade in our phylogenetic analyses. While MtKNOX8 exhibits close association with MtKNOXI proteins and demonstrates a physical

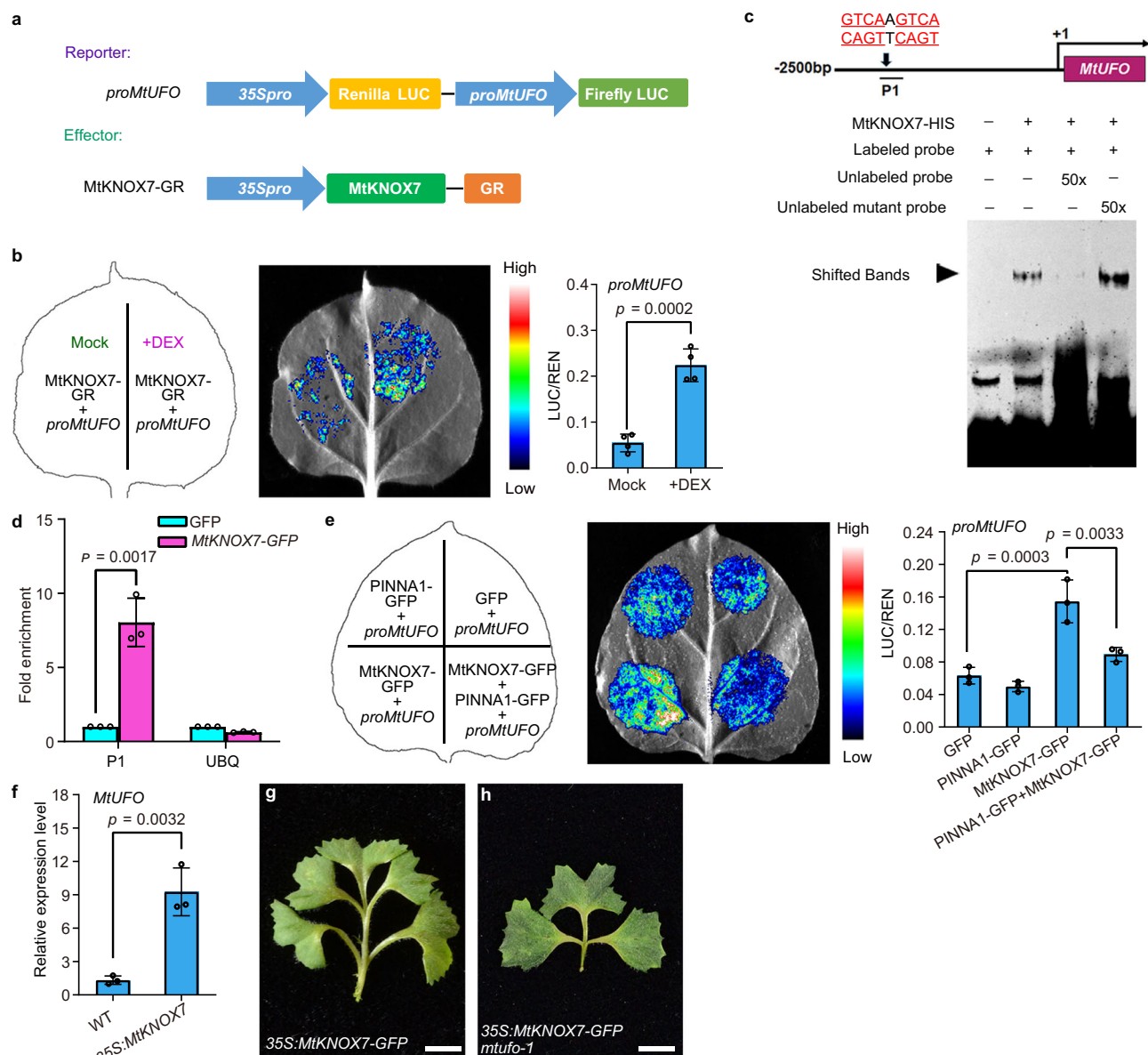

**Fig. 5 | *MtUFO* functions downstream of the PINNA1-MtKNOX7 complex.**
**a** Schematic representation of a reporter and an effector for DLR assay. The LUC (Firefly luciferase) was under the control of the *MtUFO* promoters (*proMtUFO*) as the reporter, while the REN (Renilla luciferase) was under the control of the *CaMV35S* promoter (*35Spro*) as the internal control. **b** A DLR assay indicated that the promoter of *MtUFO* was significantly activated by a GR-tagged MtKNOX7 when it was treated with dexamethasone (DEX). Data represent mean ± SD (*n* = 4 biological replicates). **c** EMSA assay showed that MtKNOX7-HIS fusion protein directly binds to the probe of the *MtUFO* promoter containing the TGACTTGAC cis-element. The arrow indicates the shifted bands. Similar results were obtained from three independent experiments. **d** ChIP-qPCR assays of the MtKNOX7-GFP protein

binding to the promoter of *MtUFO*. Data represent enrichment values normalized to *MtUBIQUITIN*. Data represent mean ± SD (*n* = 3 biological replicates). **e** A DLR assay in tobacco cells showed that the PINNA1-MtKNOX7 complex represses the MtKNOX7-mediated activation of *MtUFO* transcription. Data represent mean ± SD (*n* = 3 biological replicates). **f** The expression levels of *MtUFO* in the shoot apices of WT and *35S:MtKNOX7* were determined by qRT-PCR. *MtUBIQUITIN* was used as the internal control. Data represent mean ± SD (*n* = 3 biological replicates).
**g**, **h** Representative leaves of *35S:MtKNOX7-GFP* (**g**) and *35S:MtKNOX7-GFP mtufo-1* plants (**h**). *P* values in (**b**, **d**, **f**) were calculated by unpaired two-tailed *t*-test, while those in (**e**) were evaluated by one-way ANOVA with Tukey's multiple comparisons test. Scale bars, 5 mm in (**g**, **h**). Source data are provided as a Source Data file.

interaction with PINNA1 in *N. benthamiana* leaves (Supplementary Fig. 14), we found distinct functional roles of MtKNOX8 and MtKNOXI proteins in regulating the leaf complexity. Furthermore, MtKNOXI proteins act redundantly to modulate the leaf form in *pinna1*, but interact with PINNA1 through different functional subdomains of MtKNOXI MEINOX. These results suggest that the MtKNOX gene family can be subdivided and neofunctionalized in *M. truncatula*. Previous studies showed that alterations in the homeostasis of BLH/KNOX heterodimers can affect compound leaf development. In tomato, the mutations in *BIP* resulted in *KNOX1* overexpression and a

highly complex leaf phenotype, indicating the BIP-mediated repression of *KNOXI* activities[15]. The dominant mutant *Petroselinum* (*Pts*), characterized by a more compound leaf and higher KNOX1 expression than the wild type, was proposed to disrupt the BIP-KNOXI interaction by competing with the KNOXI protein to bind BIP[15]. In this study, the constitutive expression of *MtKNOXI* generated more leaflets, indicating that the downstream effectors were still sensitive to the MtKNOXI activity. Thus, we proposed that MtKNOXI was involved in the defects of the *pinna1* mutant in two ways. On the one hand, *PINNA1* knockout leads to the reactivation of *MtKNOXI* expression in leaf primordia,

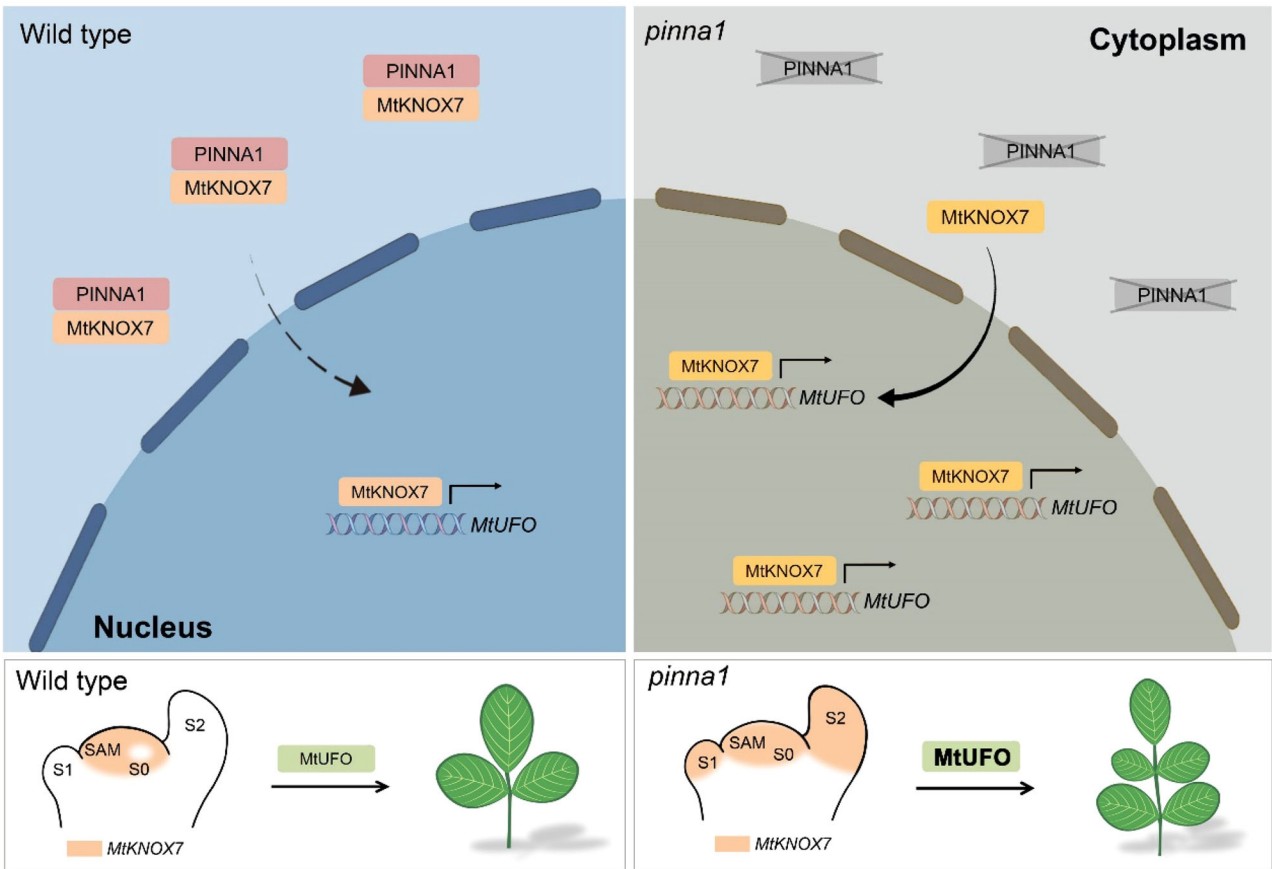

**Fig. 6 | A proposed working model for PINNA1-MtKNOX7 complex and MtUFO/MtKNOX7 module action in the compound leaf development in *M. truncatula*.** In the wild type, *PINNA1* represses the expression of *MtKNOX7*. Moreover, PINNA1 forms heterodimers with MtKNOX7 protein in the cytoplasmic space and attenuates the shuttling of MtKNOX7 from the cytoplasm into the nucleus. In *pinna1* mutant, the expression domain of *MtKNOX7* is expanded to leaf primordia, and more nuclear-localized MtKNOX7 proteins activate the transcription of *MtUFO*, thereby promoting indeterminacy in developing leaves.

whereas on the other hand, the disruption of PINNA1-MtKNOXI complexes further promotes the retention of MtKNOXI in the nucleus, allowing it to function as a transcriptional factor in the regulation of compound leaf development (Fig. 6).

The regulatory mechanism of KNOXI in leaf development has been studied in the past years. The cytokinin biosynthesis gene, *IPT7*, the target of KNOXI transcription factor STM in *Arabidopsis*[23], acts downstream of KNOXI in the regulation of compound leaf development in tomato[47]. In lettuce, LsKN1 promotes leaf complexity by directly binding to the promoters of *PINIOD*, *LsCUC3*, *LsAS1*, and *LsGA3ox1* and regulating their expression[48]. These findings suggest that KNOXI regulates leaf development in multiple ways, such as hormonal pathways and transcriptional regulatory mechanisms. *UFO* is also the possible target gene of STM during the regulation of embryogenesis and flower meristem identity in *Arabidopsis*[49,50]. Additionally, the ectopic expression of *UFO* and its homologs is able to promote indeterminacy in leaf development, which is evident in both *Arabidopsis* and *C. hirsuta*[31,32]. In this study, we identified the MtUFO/MtKNOXI module that functions in the transition from trifoliate to pinnate-like pentafoliate leaf patterns. In *M. truncatula*, the genetic and biochemical analysis revealed that the increased activity of *MtUFO* is necessary for the formation of ectopic leaflets in both *pinna1* mutants and *MtKNOX7*-overexpressing lines, and *MtUFO* was identified as one of the direct targets of MtKNOX7. Moreover, the homologs of *MtUFO*, *STP* in pea (*Pisum sativum*), and *PFO/LjUFO* in *Lotus japonicus*, were found to participate in the leaflet initiation during compound leaf development[51,52]. Thus, it might be conserved that UFO homologs play a positive role in increasing leaf complexity among legume species.

In *M. truncatula*, although the subcellular localization of PINNA1 is mainly cytoplasmic, it is also likely recruited in the nucleus through interaction with other co-factors, which might be Class II KNOX homeobox (KNOX II) or other proteins. In *Arabidopsis*, KNOXII proteins confer opposing activities with the KNOXI protein to suppress leaflet initiation during leaf development[53,54]. In *S. lycopersicum*, the members of KNOXII have also been reported to play a role in repressing leaflet formation[55]. However, whether *KNOXII* is involved in compound leaf patterning by interacting with BLH in *M. truncatula* remains an open question. Characterization of genetic relationship of the mutants with pentafoliate leaves and a comparison of regulatory targets between MtKNOXI and MtKNOXII may shed light on their roles in compound leaf patterning.

## Methods
### Plant material and growth conditions
*Medicago truncatula* ecotype R108 was used as the wild type to compare with *Tnt1* insertion mutants. The mutant lines of *PINNA1* (Medtr3g112290), *MtKNOX1* (Medtr2g024390), *MtKNOX2* (Medtr1g017080), *MtKNOX6* (Medtr5g085860), *MtKNOX7* (Medtr5g033720), *SGL1* (Medtr3g098560), and *MtUFO* (Medtr4g094748) were isolated from a *Tnt1* retrotransposon-tagged mutant collection of M. *truncatula* (Supplementary Table 1). As the *sgl1-2* and *mtufo-1* mutants were sterile, seeds were produced by the heterozygous parents for propagation. Homozygous lines were isolated from the self-pollination of heterozygous plants and genotyped in each generation. The *mtknox1-1*, *mtknox2-1*, *mtknox6-1*, *mtknox7-1* homozygous plants, *sgl1-2* heterozygous, and *mtufo-1* heterozygous

plants were crossed with *pinna1-1* homozygous lines to generate double, triple, quadruple, or quintuple mutants in the $F_1$ progeny and were identified on the basis of PCR genotyping in the $F_2$ progeny. The plants were grown in the greenhouse at 22 °C with 150 µmol m$^{-2}$ s$^{-1}$ light and 70 to 80% relative humidity under a 16-h (light)/8-h (dark) photoperiod cycle.

## Plasmid construction for transgenic plants

The coding sequence (CDS) of *MtKNOX1*, *MtKNOX2*, *MtKNOX6*, *MtKNOX7*, *MtKNOX8*, and *MtUFO* was PCR amplified and cloned into binary vectors pEarleyGate100 or pEarleyGate103-GFP to construct overexpression vectors respectively. These plasmids were introduced into the disarmed *Agrobacterium tumefaciens* strain EHA105 strain and used for plant transformation via the *Agrobacterium*-mediated transformation method to generate transgenic plants[56]. The primers used are listed in Supplementary Data 1.

## RNA extraction, RT-PCR, and qRT-PCR experiments

Total RNA was extracted from shoot apices of 6-week-old plants using RNAzol RT RNA Isolation Reagent (MRC, Cat: RN190). Plant materials were thoroughly ground using the Tissuelyser-48 (Shanghai Jingxin). Two-microgram RNA samples were subjected to reverse transcription using the All-In-One 5X RT MasterMix (abm, Cat: G592). Quantitative real-time PCR analysis was conducted on a Bio-Rad CFX Connect real-time PCR detection system (CFX96, Bio-Rad) using FastStart Universal SYBR Green Master (Rox) reagent (Roche, Cat: 4913850001). The relative transcript level was calculated using the $2^{-\Delta\Delta Ct}$ method, with *MtUBIQUITIN* (Medtr3g110110) serving as the internal control. Each reaction was conducted in triplicate. The primers used are listed in Supplementary Data 1.

## Yeast two-hybrid assay

The Matchmaker Gold Yeast Two-Hybrid System (Clontech, USA) was performed in yeast two-hybrid assays. The full-length CDS of *MtKNOX1*, *MtKNOX2*, *MtKNOX6*, *MtKNOX7*, and *MtKNOX8* were PCR amplified and cloned into the pGBKT7 vector to construct the bait plasmids (MtKNOX1-BD, MtKNOX2-BD, MtKNOX6-BD, MtKNOX7-BD, and MtKNOX8-BD), respectively. For yeast two-hybrid screening, MtKNOX7-BD was used as bait to screen a *Medicago* cDNA library. For the yeast two-hybrid assay, the full-length CDS of *PINNA1* was PCR amplified and cloned into the pGADT7 vector to construct the prey plasmid (PINNA1-AD). The bait and prey constructs were transformed into the yeast strain AH109 and grown on SD/-Trp/-Leu plates at 30 °C for 3 d. The interacting transformants were then examined on the selective medium SD/-Leu/-Trp/-His/X-a-gal/15Mm 3-AT. Protein–protein interactions were further visualized by blue color when X-α-gal was used to detect the expression of the *LacZ* reporter. The MEINOX domains of STM/BP-like KNOXI and MtKNOX7 proteins were respectively divided into three domains, MEINOX1, MEINOX2, MEINOX (MEINOX1 and MEINOX2). Truncated versions of the coding regions of *MtKNOX1*, *MtKNOX2*, *MtKNOX6*, and *MtKNOX7* were PCR amplified and cloned into the pGBKT7 vector as the bait constructs, respectively. These bait plasmids were then co-transformed with PINNA1-AD into the yeast strain. The subsequent steps followed the procedure described above. The primers used are listed in Supplementary Data 1.

## Bimolecular fluorescence complementation (BiFC) assay

The full-length CDS of *MtKNOX1*, *MtKNOX2*, *MtKNOX6*, and *MtKNOX7* were individually cloned into the pEarly201-ccdB-nYFP (YN) vector, while the full-length CDS of PINNA1 was cloned into the pEarly202-ccdB-cYFP (YC) vector. Additionally, the full-length CDS of *ATG8e* and *NBR1* were cloned into the pCAMBIA3300-mCherry vector to serve as markers for autophagosomes or autophagic bodies. Subsequently, the corresponding binary vectors were introduced into *Agrobacterium*

*tumefaciens* strain GV3101 and co-infiltrated into the leaves of *Nicotiana benthamiana* to test the interaction. The fluorescence signal was monitored using a Zeiss LSM880 confocal laser scanning microscope (Zeiss, Germany). All primers used are listed in Supplementary Data 1.

## Coimmunoprecipitation (co-IP) assay

The full-length CDS of *PINNA1*, *MtKNOX1*, *MtKNOX2*, *MtKNOX6*, and *MtKNOX7* were PCR amplified and cloned into pENTR/D-TOPO cloning vector (Invitrogen, Cat: K240020). Following this, they were transformed into either the pCAMBIA1390-ccdB-7Myc-6His or pEarleyGate103-GFP vector using the Gateway LR Clonase II (Invitrogen, Cat: 11791020). These constructs were expressed individually or co-expressed in the leaves of 4-week-old *N. benthamiana* plants after *Agrobacterium* infiltration. Equal amounts of samples (0.3 g) were collected after a 48-h incubation and powdered in liquid nitrogen. Total protein was extracted with 1 mL extraction buffer (20 mM HEPES-KOH (pH 7.5), 150 mM NaCl, 1 mM EDTA, 0.5% Triton X-100, 0.2% β-mercaptoethanol, 10% glycerol, 1 mM phenylmethylsulfonyl fluoride (PMSF), and 1x protease inhibitor cocktail). Samples were incubated at 4 °C for 20 min with gentle shaking and centrifuged twice at 15,000×*g* for 20 min at 4 °C. The supernatant was incubated with 10 µL of GFP-Trap magnetic agarose beads (Chromotek, Cat: gtma-20) at 4 °C for 4 h by vertical rotation. Subsequently, the beads were washed seven times with extraction buffer and eluted with 2× SDS protein sample loading buffer for 5 min at 95 °C. The supernatants were electrophoretically separated by 10% SDS-PAGE and transferred to a nitrocellulose membrane (Bio-Rad, Cat: 1620115). Immunoblots analysis was performed using a mouse anti-Myc antibody (ABclonal, Cat: AE010, 1:4000 dilution) and a mouse anti-GFP antibody (TransGen Biotech, Cat: HT801-01, 1:4000 dilution). The primers used are listed in Supplementary Data 1.

## Subcellular localization assay

For subcellular localization in *M. truncatula* mesophyll protoplast, full-length CDS of *PINNA1*, *MtKNOX1*, *MtKNOX2*, *MtKNOX6*, and *MtKNOX7* were PCR amplified and cloned into PBI221-GFP or PAN583-mCherry vectors respectively. These constructs were transiently expressed alone or co-expressed in *Medicago* mesophyll protoplasts using the polyethyleneglycol (PEG)–mediated transformation method[57]. The infiltrated protoplasts were incubated at 22 °C for 12–16 h in the dark. For subcellular localization in *Medicago* leaf epidermal cells, PINNA1-GFP, GFP-PINNA1, and 35S-GFP constructs were transiently expressed in leaves by *Agrobacterium* infiltration. The fluorescence signal was monitored by a Zeiss LSM880 confocal laser scanning microscope (Zeiss, Germany). The primer sequences are listed in Supplementary Data 1.

## Scanning electron microscopy (SEM) assay

For SEM, shoot apices samples from 4-to 6-week-old plants were fixed in a fixative solution (3% glutaraldehyde in 1×PBS buffer) under vacuum for 10 min at 4 °C, followed by overnight incubation at 4 °C. The samples were then washed five times in 1×PBS buffer, with each wash lasting 10 min. Subsequently, the samples were dehydrated in graded ethanol (30, 50, 60, 70, 85, 95, and two rounds of 100% ethanol, each for 20 min). After dehydration, the samples underwent critical point drying by carbon dioxide and were then sprayed with gold powder. Finally, the prepared samples were observed by Quanta 250 FEG scanning electron microscope (FEI, USA) at an accelerating voltage of 5 kV.

## RNA in situ hybridization

The RNA in situ hybridization treatments were conducted following the previously established methods[58]. Briefly, 777-bp, 518-bp and 640-bp fragments were isolated and PCR amplified from the CDS of *MtKNOX7*, *PINNA1*, and *MtUFO*, respectively. The resulting PCR

products were ligated into the pGEM-T vector (Promega, Cat: A3600) using the primer sequences provided in Supplementary Data 1. Both the sense and antisense probes were labeled with digoxigenin (Roche, Cat: 11175025910) according to the manufacturer's instructions. RNA in situ hybridization was implemented on shoot apices of 4-week-old wild-type and *pinna1-6* plants with the digoxigenin-labeled sense or antisense probes. The fresh shoot apices were fixed in FAA solution at 4 °C and then embedded in paraffin (Leica, Cat: 39601095). After the sectioning and hybridization processes, the 8 μm-thick slices of shoot-apex samples were examined using ECLIPSE Ni Series light microscope (Nikon, Japan).

### Nuclear-cytoplasmic fractionation assay

For nuclear-cytoplasmic fractionation, the full-length CDS of *PINNA1* was PCR amplified and cloned into PAN583-GFP or PAN583-4Myc vectors respectively. The primer sequences are listed in Supplementary Data 1. The nuclear and cytoplasmic fractions were isolated from protoplasts expressing GFP- or Myc-tagged proteins, following the previously established protocol[59]. *Medicago* mesophyll protoplast transformed with plasmids were cultured for 16 h at 22 °C under dark conditions, then resuspended in 1 mL extraction buffer (20 mM Tris-HCl pH 7.5, 250 mM sucrose, 25% glycerol, 20 mM KCl, 2 mM EDTA, 2.5 mM MgCl$_2$, 5 mM DTT, 1× protease inhibitor cocktail, and 0.3% Triton X-100). After incubating on ice for 10–15 min, 100 μL of the mixture was taken as the total protein fraction. The crude cytoplasmic and nuclear fractions were separated by centrifugation at 3000×g for 10 min at 4 °C. The supernatant was centrifuged at 12,000×g for 10 min at 4 °C and collected as the cytoplasmic fraction. The pellet was washed three times at 3000 × g for 5 min with 1 mL of nuclear wash buffer (20 mM Tris-HCl, pH 7.4, 25% glycerol, 2.5 mM MgCl$_2$, 0.1% Triton X-100, and 1× protease inhibitor cocktail). Subsequently, it was resuspended in with 500 μL of NRB2 buffer (20 mM Tris-HCl, pH 7.5, 0.25 M Sucrose, 10 mM MgCl$_2$, 0.5% Triton X-100, 5 mM β-mercaptoethanol, and 1× protease inhibitor cocktail) and carefully overlaid on top of 500 μL NRB3 buffer (20 mM Tris-HCl, pH 7.5, 1.7 M Sucrose, 10 mM MgCl$_2$, 0.5% Triton X-100, 5 mM β-mercaptoethanol and 1× protease inhibitor cocktail). These were centrifuged at 16,000 × g for 30 min at 4 °C. The final nuclear pellet was resuspended in 100 μL extraction buffer. To ensure the quality of the fractions, PEPC protein and Histone H3 were used as cytoplasmic or nuclear markers. Immunoblots analysis was performed using a mouse anti-Myc antibody (ABclonal, Cat: AE010, 1:4000 dilution), a mouse anti-GFP antibody (TransGen Biotech, Cat: HT801-01, 1:4000 dilution), a rabbit anti-Histone-H3 antibody (Proteintech, Cat: 17168-1-AP, 1:2000 dilution), and a rabbit anti-PEPC antibody (PhytoAB, Cat: PHY2038S, 1:2000 dilution).

### Transcriptional activation assay

The transcriptional activation assay was conducted in *Arabidopsis* mesophyll protoplasts using a dual-luciferase reporter (DLR) assay system (Promega, Cat: E1960). The reporter *6×UASpro* pGREENII vector contains six GAL4 binding elements (upstream activating sequence, UAS), a Renilla luciferase (REN) gene (as an internal control), and a Firefly luciferase (LUC) gene (as a reporter). For effectors, coding regions of GAL4BD-MtKNOXIs protein fusion were amplified using specific primers from the GAL4BD fusion plasmids of yeast two-hybrid assays and cloned into PBI221 to construct effector plasmids. The coding region of PINNA1 was also amplified and cloned into PBI221 as another effector. The primer sequences are listed in Supplementary Data 1. These constructs were transiently expressed in *Arabidopsis* protoplasts. The infiltrated protoplasts were kept in the dark for 12–16 h, and the relative LUC and REN activity was measured by the DLR assay system on the Centro XS LB960 (Berthold, Germany). The plasmid ratio for the experimental setup was *6×UASpro* pGREENII: BD-MtKNOXI-PBI221: PINNA1-PBI221 = 1: 4: 4 (μg: μg: μg), compared

with the control (*6XUASpro* pGREENII: BD-MtKNOXI-PBI221: PBI221 = 1: 4: 4). Finally, the activity of transactivation was indicated by the ratio of LUC to REN.

### Luciferase imaging assay

The full-length CDS of *PINNA1*, *MtKNOX1*, *MtKNOX2*, *MtKNOX6*, and *MtKNOX7* were PCR amplified and fused to either the pCAMBIA3300-GFP or pCAMBIA3300-GR vector to generate effectors, respectively. The ~2.9-kb promoter fragment upstream of *MtUFO* was cloned into the pGreenII-0800-Luc vector to generate a reporter construct. The constructs were introduced into *A. tumefaciens* GV3101 together with the *pSoup19* helper plasmid. At least four tobacco leaves from independent plants were infiltrated and harvested after 48 h. For GR induction, another set of at least four tobacco leaves from independent plants were infiltrated with infiltration buffer containing 10 μM DEX (Coolaber, Cat: SL4070) or an equal volume of ethanol, and harvested after 48 h. The luciferase fluorescence signals were observed by a plant living imaging system (Tanon 5200, China) after these leaves were sprayed with 1 mM D-Luciferin solution (GoldBio, Cat: LUCK-1G) and incubated for 5 min in the dark. Then the rest of the leaves were punched and powdered in liquid nitrogen to measure the LUC and REN activity. Relative LUC and REN activity were measured by the DLR assay system on the Centro XS LB960 (Berthold, Germany). Finally, the relative firefly luciferase activity was calculated as the ratio of LUC to REN for each sample. The primer sequences are listed in Supplementary Data 1.

### Electrophoretic mobility shift (EMSA) assay

To express MtKNOX7 in *E. coli*, the full-length CDS of *MtKNOX7* was cloned into the pET28a vector. The MtKNOX7-HIS fusion protein was expressed in *E. coli* Rosetta (DE3) cells (TransGen Biotech, Cat: CD801-02) by induction with 0.5 mM isopropyl β-D-thiogalactoside (IPTG) at 16 °C for overnight, and then purified with Ni Sepharose 6 Fast Flow (GE Healthcare, Cat: 17531803). For the EMSA assay, two complementary oligonucleotides with 5' biotin labeling were synthesized and then hybridized to prepare double-strand DNA probes. The EMSA assay was performed using the LightShift Chemiluminescent EMSA Kit (Thermo Fisher Scientific, Cat: 20148) following the manufacturer's instructions. The primer sequences are listed in Supplementary Data 1.

### Chromatin immunoprecipitation (ChIP) assay

For ChIP assays[60], plant tissues (1 g) were cross-linked in cross-linking buffer (10 mM Tris-HCl pH = 8, 1 mM PMSF, 1 mM EDTA 0.4 M sucrose, and 1% formaldehyde) for 10 min and then quenched the crosslinking reaction with Glycine (0.25 M) for 5 min at room temperature. After being powdered in liquid nitrogen, the pellet was washed twice with Nuclei Extraction Buffer (15 mM PIPES pH 6.8, 5 mM MgCl$_2$, 50 mM KCl, 15 mM NaCl, 1 mM CaCl$_2$, 0.25 M sucrose, 0.4 % Triton X-100, 5 mM PMSF, and 1x protease inhibitor cocktail), and then centrifuged at 6000 × g at 4 °C for 5 min to isolate nuclei. The pelleted nuclei were resuspended in Nuclear Lysis Buffer (50 mM Tris-HCl (pH 8.0), 10 mM EDTA (pH 8.0), 1% SDS, 1 mM PMSF, and 1x protease inhibitor cocktail) and sonicated with Bioruptor to shear the DNA into fragments of 200 to 500 bp. Immuno-precipitate chromatin complexes were incubated with anti-GFP antibody (Abcam, Cat: ab290) at 4 °C overnight, and subsequently mixed with protein A/G magnetic beads (MCE, Cat: HY-K0202) at 4 °C for 2 h. DNA was eluted and purified from the complexes, and the enrichment of DNA fragments was analyzed by quantitative PCR using the primer listed in Supplementary Data 1.

### Phylogenetic analysis

To conduct the phylogenetic analysis, KNOX homologs or UFO orthologs were retrieved from PLAZA for most species, respectively.

Multiple sequences were aligned using ClustalW online (https://www.genome.jp/tools-bin/clustalw) with default parameters. The neighbor-joining phylogenetic tree was built using the MEGA11 program (http://www.megasoftware.net/), and the phylogenetic trees with bootstrap values from 1,000 trials were shown. All valid KNOX homologs were listed in Supplementary Data 2, 3, while Supplementary Data 4 included all valid UFO orthologs.

## Statistical analysis

Statistical analysis was conducted using the program GraphPad Prism version 9.0 software. An unpaired two-tailed *t*-test was performed to calculate the *P* values between two groups, while for comparisons involving three or more groups, a one-way ANOVA with Tukey's multiple comparisons test was conducted. The results of the statistical analyses are shown in the Source Data.

## Accession numbers

Sequence data from this article can be found in GenBank under the following accession numbers: *PINNA1* (Medtr3g112290), *MtKNOX1* (Medtr2g024390), *MtKNOX2* (Medtr1g017080), *MtKNOX6* (Medtr5g085860), *MtKNOX7* (Medtr5g033720), *MtKNOX8* (Medtr1g084060), *SGL1* (Medtr3g098560), *MtUFO* (Medtr4g094748), *MtActin* (Medtr7g026230), *MtUBIQUITIN* (Medtr3g110110), *NBR1*(AT4G24690), and *ATG8e* (AT2G45170).

## Reporting summary

Further information on research design is available in the Nature Portfolio Reporting Summary linked to this article.

## Data availability

All data in this study are available in the manuscript or the Supplementary materials. Source data are provided with this paper.

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

## Acknowledgements

We thank Prof. Kirankumar Mysore (Oklahoma State University) and Dr. Jiangqi Wen (Oklahoma State University) for providing the *Tnt1* mutants; Prof. Mingyi Bai (Shandong University) for providing the plasmids for co-IP assay; members of F.X. and Z.C. laboratory for providing their valuable input; Haiyan Yu, Yuyu Guo, and Xiaomin Zhao from the Analysis and Testing Center of SKLMT (State Key Laboratory of Microbial Technology, Shandong University) for assistance with the laser scanning confocal microscopy. This work was supported by grants from the National Key Research and Development Program of China (2023YFF1001400), the National Natural Science Foundation of China (32370881, 32170833, 31900172, and 32201446), and Shandong Province (ZR2020KC018).

## Author contributions

Z.L. and C.Z. designed research; Z.L., J.J.Z., H.W., K.Z., Y.X., and J.Z. performed research; Z.G., L.H., and F.X. contributed new reagents/analytic tools; Z.L., J.J.Z., H.W., Z.G., and M.W. analyzed data; and Z.L and C.Z. wrote the paper.

## Competing interests

The authors declare no competing interests.
