## [Peer Review File · Nature Communications]

REVIEWER COMMENTS

Reviewer #1 (Remarks to the Author):

The paper titled "Rewiring of a KNOX1 regulatory network mediated by UFO underlies the compound leaf development in *Medicago truncatula*" authored by Zhichao Lu et al provides a detailed exploration of the regulatory mechanisms behind compound leaf development in *M. truncatula*. Investigating the mechanisms of compound leaf development holds great significance in the field of evolutionary developmental biology in plants. While previous studies primarily focused on tomato, revealing the association between ectopic expression of KNOX1 in leaf primordia and increased leaf complexity, this paper highlights an intriguing departure in legume species, where KNOX1 genes are not implicated in compound leaf development. Through genetic and molecular analyses, the authors uncover a novel pathway that represses KNOX1 function in *M. truncatula*.

The experiments and analyses are carefully performed and the conclusions are reasonable. The findings of this paper would attract a wide range of readers, especially in the field of evolutionary developmental biology. In conclusion, the paper would be sufficient to merit publication in Nature Communications though a revision is recommended which needs to include the following points.

(1) In page 3, line 86, "KNOX1" should be italic

(2) In Figure 1 and page 5, line 125, the authors mention that MtKNOX8 belongs to KNOX1 family. However, we cannot say that it belongs to KNOX1 based on the phylogenetic tree only. It would be appreciated to add the multiple sequence alignment (clustalw) in supplemental data and an outgroup gene to make sure it belongs to KNOX1.

(3) In Figure S1, it is better to show how many samples are calculated in this experiment. The authors claim that they divide these overexpressing plants into weak and strong phenotypes by their expression levels (page 5, line 134), so the expression of these plants should be shown in each part. In addition, quantifying their leaf complexity would be appreciated to compare the difference (doi: 10.1093/plphys/kiac382).

(4) It is very interesting that MtKNOX1/2/6/7 interact with PINNA1 physically and is verified by BIFC. I'd like to know which organelle function in tobacco leaves. It seems that they accumulated in a small region of cytoplasm (Figure 2A), so what part is it? If it is difficult to perform additional experiments, an explanation would be appreciated.

(5) In FigureS 7b, the author should add control of nucleus by DAPI or Hoechst staining.

(6) What is the expression pattern of MtUFO in leaf primordia, does it show a similar pattern with MtKNOX7 in figure3?

(7) Figure 2I is too small. Please show LL in the figure and clearly indicate that the location where the ELL is formed is at the base of the TL.

(8) MtKNOX2 and MtKNOX6 GFP localization appear to be mostly in the nucleus as seen in Fig. 3A. The results of Fig. 3B give a different impression. Is there really a big difference in the localization of MtKNOX2,6 and MtKNOX7 and can explain the difference in regulatory mechanisms? If typeII localization accounts for more than 80% in FigS7D, it would be better to use typeII pictures in Fig3A.

(9) in Fig5 E, It is difficult to see "PINNA1-GFP+ MtKNOX7-GFP" in the bar graph. The space between the lines should be narrower.

Reviewer #2 (Remarks to the Author):

This paper investigates the role of various regulators on leaf development in Medicago. pinna mutants make 5 leaflets while normal Medicago has 3. This group shows that knockouts of all combinations of knox genes has no phenotype (still 3 leaflets) but overexpression of these KNOX genes leads to ectopic leaflets.

They determine that PINNA interacts with KNOX genes 1, 2, 6 and 7 and sequesters the KNOX into the cytoplasm. PINNA by itself is in the cytoplasm. This is interesting because in other species PINNA-KNOX heterodimers are in the nucleus. These results are well substantiated by a number of methods, including nuclear-cytoplasmic fractionation. One suggestion could be is to include the coimmunoprecipitation of KNOX8 and PINNA, which did not interact in yeast two hybrid as a control. It is slightly unnerving to look at a figure and see all the panels showing the same result.

They also show that making knox;pinna double mutants restores the phenotype to a normal 3 leaflets to some degree with the best restoration when all knox genes are missing. They also showed by RNA in situ (which are done well) that KNOX7 and PINNA overlap at the boundary of leaf initiation and that in the pinna mutant, the KNOX expression spreads into the leaflet. Thus PINNA keeps KNOX out of the nucleus so that it can't activate additional leaflet formation.

A major regulator of leaflet formation is SGL1 (ortholog of LFY). The *sgl1* mutant phenotype is no leaflets at all. The *pinna;sgl1* double mutant like *sgl1* (but occasional leaves have an extra leaflet or two) leaflets, thus *sgl1* is epistatic to *pinna*. In a previous report this group showed that overexpression of SGL1 does not have more leaflets. (by the way, do they know that more protein or more transcript is produced in those overexpressors?).

In other systems LFY functions with UFO. They show that overexpression of UFO gives more leaflets and it looks like *pinna*. They make the Medicago loss of function of UFO and while it has no phenotype on its own, the double with *pinna*, has 3 leaflets. So, the 5 leaflets in *pinna* mutants require a functional UFO.

Here they bring KNOX back in. They show that it is KNOX7 that binds and regulates the UFO promoter (not PINNA) and when PINNA is included in the assay, the upregulation is decreased. They also show that the extra leaflets of 35S:KNOX7 are reduced when combined with loss of function *ufo*. Thus KNOX7 has the capacity to activate UFO to make extra leaflets, but PINNA keeps KNOX from entering the nucleus and turning on UFO.

They end the results by exploring KNOX1, 2, 6 to see if they are similar to KNOX7. I am not sure they need this last bit, but it is not up to me. I am happy seeing clear results with KNOX7.

Figure 2L. Is the last lane mislabeled? I thought *knox* LOF were always just like normal. Should this be labeled *pinna knox6 knox7*?

Figure 3C labeling. They need to put the two genes that are coexpressed closer together.

Figure 5E labeling is also tricky.

Dear Reviewers,

Thank you very much for the valuable comments. We have carefully studied the comments and revised the manuscript accordingly. Please find below point to point response to the comments/suggestions. All the changes are marked in red color in the revised manuscript.

Response to Reviewer #1

Reviewer #1 (Remarks to the Author):

1. The paper titled "Rewiring of a KNOXI regulatory network mediated by UFO underlies the compound leaf development in *Medicago truncatula*" authored by Zhichao Lu et al provides a detailed exploration of the regulatory mechanisms behind compound leaf development in *M. truncatula*. Investigating the mechanisms of compound leaf development holds great significance in the field of evolutionary developmental biology in plants. While previous studies primarily focused on tomato, revealing the association between ectopic expression of KNOXI in leaf primordia and increased leaf complexity, this paper highlights an intriguing departure in legume species, where KNOXI genes are not implicated in compound leaf development. Through genetic and molecular analyses, the authors uncover a novel pathway that represses KNOXI function in *M. truncatula*. The experiments and analyses are carefully performed and the conclusions are reasonable. The findings of this paper would attract a wide range of readers, especially in the field of evolutionary developmental biology. In conclusion, the paper would be sufficient to merit publication in Nature Communications though a revision is recommended which needs to include the following points.

Response: Thank you for your critical reading and suggestions. Our responses to all comments and questions are listed below.

2. In page 3, line 86, "KNOX1" should be italic

Response: Thank you for pointing this. We have corrected these types of font error in the manuscript as you mentioned.

3. In Figure 1 and page5, line125, the authors mention that MtKNOX8 belongs to KNOX1 family. However, we cannot say that it belongs to KNOX1 based on the phylogenetic tree only. It would be appreciated to add the multiple sequence alignment (clustalw) in supplemental data and an outgroup gene to make sure it belongs to KNOX1.

Response: Thank you for your suggestions. To make it clear, we conducted a multiple sequence alignment (clustalw) of Class I KNOX proteins in *M. truncatula* and *A.thaliana*. The new Supplementary Fig.1 shows that the N-terminus region of MtKNOX8 has a shorter amino acid sequence (N-terminus region) compared to other KNOXI proteins. This result implies that the function of MtKNOX8 is different from other members of the MtKNOXI proteins. Additionally, the revised Supplementary Fig. 2a shows that MtKNOX8 did not form a cluster with any other MtKNOXI proteins. Therefore, we think that MtKNOX8 is a member of MtKNOXI subfamily, but has different function due to the difference in amino acid sequence.

We have added the result of clustalw analysis in the manuscripts (page 5).

4. In Figure S1, it is better to show how many samples are calculated in this experiment. The authors claim that they divide these overexpressing plants into weak and strong phenotypes by their expression levels (page 5, line 134), so the expression of these plants should be shown in each part. In addition, quantifying their leaf complexity would be appreciated to compare the difference (doi: 10.1093/plphys/kiac382).

Response: Thank you for your suggestions. We have added and labeled the statistics data regarding the sample size of overexpressing plants in the revised Fig. 1c-g and the revised Supplementary Fig. 2c-f. Furthermore, we have added data of expression levels of *MtKNOXI* in the overexpressing plants in Supplementary Fig. 2. The overexpressing plants with strong phenotypes displayed significantly higher expression levels of *MtKNOXI* compared to the overexpressing plants with weak phenotypes.

5. It is very interesting that MtKNOX1/2/6/7 interact with PINNA1 physically and is verified by BiFC. I'd like to know which organelle function in tobacco leaves. It seems that they accumulated in a small region of cytoplasm (Figure2A), so what part is it? If it is difficult to perform additional experiments, an explanation would be appreciated.

Response: Thank you for these questions. In order to determine the organelle localization of the BiFC complexes, we employed various organelle markers. Notably, we observed that the BiFC signals co-localized with the puncta of the autophagosome marker mCherry-ATG8 and the autophagy receptor NBR1-mCherry (see the new Supplementary Fig. 4a and b). This observation indicates that the complexes were localized within autophagosomes or autophagic bodies, suggesting that they have the potential to undergo degradation through the autophagy process in tobacco leaves. We have added these results in the manuscripts (page6 to page7).

6. In Figure S 7b, the author should add control of nucleus by DAPI or Hoechst staining.

Response: Thank you for your critical reading and suggestions. We made several attempts to stain the nuclei of *M. truncatula* leaf cells using DAPI or Hoechst 33342, but unfortunately, all our efforts were unsuccessful. The cell walls of *M. truncatula* leaves are easily stained by these kinds of dyes, which interfered with the specific staining of the nuclei. To solve this problem, we conducted transient co-expression experiments of PINNA1-GFP, GFP-PINNA1, or 35S-GFP with the nuclear marker H2B-mCherry in the epidermal cells of *M. truncatula* leaves. Similarly, we observed that PINNA1-GFP and GFP-PINNA1 did not co-localize with H2B-mCherry, indicating that PINNA1 was not localized to the nucleus. We added these data in revised Supplementary Fig.8b and in manuscripts (page9).

7. What is the expression pattern of MtUFO in leaf primordia, does it show a similar pattern with MtKNOX7 in figure3?

Response: Thank you for these questions. We perform the RNA in situ hybridization experiments to check the expression pattern of *MtUFO* in the leaf primordia between WT and *pinna1-6* mutant. In WT, the spatial localization of *MtUFO* was primarily observed at the boundary between the shoot apical meristem (SAM) and the leaf primordia (Supplementary Fig. 11d), which is partially overlapped with that of *MtKNOX7* (Fig. 3e). However, the spatial localization of *MtUFO* was expanded and showed ectopic presence in the leaf primordium of *pinna1-6* mutant (Supplementary Fig. 11e). Notably, the ectopic expression of *MtUFO* was also overlapped with the ectopic expression of *MtKNOX7* in the leaf primordium of *pinna1-6* mutant at S1 stage.

We added these data in revised Supplementary Fig. 11d-e, and in the manuscripts (line 447-450).

8. Figure 2I is too small. Please show LL in the figure and clearly indicate that the location where the ELL is formed is at the base of the TL.

Response: Thank you for pointing this. Due to the space restriction in the new Fig.2, we moved these SEM images into the Supplementary Fig. 6e-h. In order to enhance readability, we have proceeded to enlarge the SEM images and incorporate clear labels (see the new Supplementary Fig.6e-h).

9. MtKNOX2 and MtKNOX6 GFP localization appear to be mostly in the nucleus as seen in Fig. 3A. The results of Fig. 3B give a different impression. Is there really a big difference in the localization of MtKNOX2,6 and MtKNOX7 and can explain the difference in regulatory mechanisms? If typeII localization accounts for more than 80%

in FigS7D, it would be better to use typeII pictures in Fig3A.

Response: Thank you for your questions. We are very certain that there really a big difference in the localization of MtKNOX1,2,6 and MtKNOX7 from multiple biological replicates. Based on our data, we found that less than 10% of MtKNOX1/2/6-GFP exhibited nucleocytoplasmic distribution, while the majority predominantly exhibited cytoplasmic distribution. In contrast, all MtKNOX7-GFP samples exhibited nucleocytoplasmic distribution. Currently, the specific differences in regulatory mechanisms governing the localization of MtKNOX1,2,6 and MtKNOX7 remain unclear. In the discussion section of our manuscripts (page 14), we have discussed the potential regulatory mechanisms that may account for the difference in the localization of MtKNOX1,2,6 and MtKNOX7. In rice, three KNOXI (Oskn1/OSH1, Oskn2/OSH71, and Oskn3/OSH15) proteins showed different nuclear and cytoplasmic localization patterns (Kuijt et al., 2004). Specifically, Oskn2 and Oskn3 primarily localize to the nucleus and cytoplasm, whereas Oskn1 exhibits either cytoplasmic or nucleocytoplasmic distribution. Additionally, it is worth noting that the subcellular localization of Oskn1/2/3 can be influenced by plant hormones (Kuijt et al., 2004). Moreover, the presence of other proteins, such as members of the MCTP family and microtubule-associated proteins, can also impact the subcellular localization of KNOXI members (Winter et al., 2007; Liu et al., 2018; Song et al., 2018). Therefore, the diverse subcellular localizations observed between MtKNOX1/2/6 and MtKNOX7 within individual cells may be attributed to the involvement of multiple regulatory mechanisms. These mechanisms might include protein-protein interactions with members of the MCTP family or microtubule-associated proteins, as well as potential influences from plant hormones or other cellular components.

To make this point clear, we have simultaneously added pictures of typeI and typeII in Fig. 3a based on your suggestion. Thank you for your suggestions again.

10. in Fig5 E, It is difficult to see “PINNA1-GFP+ MtKNOX7-GFP” in the bar graph. The space between the lines should be narrower.

Response: Thank you for pointing this. We have modified the bar graph to improve their readability (see the new Fig.5e).

Response to Reviewer #2

Reviewer #2 (Remarks to the Author):

1. This paper investigates the role of various regulators on leaf development in *Medicago*. pinna mutants make 5 leaflets while normal *Medicago* has 3. This group shows that knockouts of all combinations of knox genes has no phenotype (still 3 leaflets) but overexpression of these KNOX genes leads to ectopic leaflets. They

determine that PINNA interacts with KNOX genes 1, 2, 6 and 7 and sequesters the KNOX into the cytoplasm. PINNA by itself is in the cytoplasm. This is interesting because in other species PINNA-KNOX heterodimers are in the nucleus. These results are well substantiated by a number of methods, including nuclear-cytoplasmic fractionation. One suggestion could be to include the coimmunoprecipitation of KNOX8 and PINNA, which did not interact in yeast two hybrid as a control. It is slightly unnerving to look at a figure and see all the panels showing the same result.

Response: Thank you for the comments and suggestions. Follow your suggestion, we have performed coimmunoprecipitation assay and BiFC assay between KNOX8 and PINNA1. Different from the result of the yeast two-hybrid assay, we found that PINNA1 could interact with KNOX8 in tobacco leaves based on the result of co-IP assay and BiFC assay (see the new Supplementary Fig.14). However, our data showed that ectopic expression of MtKNOX8 did not result in an increase in leaf complexity, unlike the ectopic expression of *MtKNOX1/2/6/7*. Therefore, MtKNOX8 is not responsible for the phenotype of *pinna1*. We discussed this point in page 15.

2. They also show that making *knox;pinna* double mutants restores the phenotype to a normal 3 leaflets to some degree with the best restoration when all *knox* genes are missing. They also showed by RNA in situ (which are done well) that KNOX7 and PINNA overlap at the boundary of leaf initiation and that in the *pinna* mutant, the KNOX expression spreads into the leaflet. Thus PINNA keeps KNOX out of the nucleus so that it can't activate additional leaflet formation. A major regulator of leaflet formation is SGL1 (ortholog of LFY). The *sgl1* mutant phenotype is no leaflets at all. The *pinna;sgl1* double mutant like *sgl1* (but occasional leaves have an extra leaflet or two) leaflets, thus *sgl1* is epistatic to *pinna*. In a previous report this group showed that overexpression of SGL1 does not have more leaflets. (by the way, do they know that more protein or more transcript is produced in those overexpressors?).

Response: Thank you for the comments and questions. To make it clear, we analyzed the expression levels of *SGL1* in the shoot apices of WT and *35S:SGL1* plants. The data showed that the expression levels of *SGL1* in *35S:SGL1* plants were 112-fold higher than those observed in WT plants (see the new Supplementary Fig.10h).

3. In other systems LFY functions with UFO. They show that overexpression of UFO gives more leaflets and it looks like *pinna*. They make the *Medicago* loss of function of UFO and while it has no phenotype on its own, the double with *pinna*, has 3 leaflets. So, the 5 leaflets in *pinna* mutants require a functional UFO. Here they bring KNOX back in. They show that it is KNOX7 that binds and regulates the UFO promoter (not PINNA) and when PINNA is included in the assay, the upregulation is decreased. They also show that the extra leaflets of *35S:KNOX7* are reduced when combined with loss of function *ufo*. Thus KNOX7 has the capacity to activate UFO to make extra leaflets, but PINNA keeps KNOX from entering the nucleus and turning

on UFO. They end the results by exploring KNOX1, 2, 6 to see if they are similar to KNOX7. I am not sure they need this last bit, but it is not up to me. I am happy seeing clear results with KNOX7.

Response: Thank you for the comments and suggestions. Follow your suggestion, we have now removed the last bit that you mentioned from our manuscripts.

4. Figure 2L. Is the last lane mislabeled? I thought knox LOF were always just like normal. Should this be labeled pinna knox6 knox7?

Response: Thank you for pointing this. We revised the label in Fig.2h, and the last line was revised to *pinna1-1 mtknox1-1 mtknox2-1 mtknox6-1 mtknox7-1*. As shown in Fig.2h, knockout of all MtKNOX1/2/6/7 genes in *pinna1-6* rescued 79.8% leaves of the mutant.

5. Figure 3C labeling. They need to put the two genes that are coexpressed closer together.

Response: Thank you for pointing this. We have modified the graph to improve their readability (see Fig.3c)

6. Figure 5E labeling is also tricky.

Response: Thank you for pointing this. We have modified the graph to improve their readability (see Fig.5e)

REVIEWERS' COMMENTS

Reviewer #1 (Remarks to the Author):

The authors adequately addressed all of my comments.

Reviewer #2 (Remarks to the Author):

I am happy with the changes to the manuscript and think they have done an excellent job. Below are some minor points.

Minor point, but line 65. You have clearly stated that you are talking about TALE proteins in plants, so you probably don't need the beginning of the sentence "In plants,..."

Line 107.why say possible target? Don't you show it is a target?

line 373 maybe replace "which is hard to regulate the transcription" with "which inhibits transcriptional regulation"

REVIEWERS' COMMENTS

Reviewer #1 (Remarks to the Author):

The authors adequately addressed all of my comments.

Response: Thanks for your careful review.

Reviewer #2 (Remarks to the Author):

I am happy with the changes to the manuscript and think they have done an excellent job. Below are some minor points.

Response: Thank you for your appreciation of our manuscript and the careful review and suggestions. Following your suggestions, we have modified some of the sentences in the manuscript that you pointed out.

Minor point, but line 65. You have clearly stated that you are talking about TALE proteins in plants, so you probably don't need the beginning of the sentence "In plants, ...

Response: Thank you for pointing this out. We have deleted "In plants" from this sentence.

Line 107. Why say possible target? Don't you show it is a target?

Response: Thank you for your critical reading. We have modified this sentence and deleted the word "possible".

line 373 may replace "which is hard to regulate the transcription" with "which inhibits transcriptional regulation"

Response: Thank you for pointing this out and providing your suggestion. Following your suggestion, we have replaced "which is hard to regulate the transcription" with "which inhibits transcriptional regulation" in this sentence.